# An Investigation of the Effects of Brain Fatigue on the Sustained Attention of Intelligent Coal Mine VDT Operators

**DOI:** 10.3390/ijerph191711034

**Published:** 2022-09-03

**Authors:** Linhui Sun, Zigu Guo, Xiaofang Yuan, Xinping Wang, Chang Su, Jiali Jiang, Xun Li

**Affiliations:** 1School of Management, Xi’an University of Science and Technology, Xi’an 710054, China; 2Research Center for Human Factors and Management Ergonomics, Xi’an University of Science and Technology, Xi’an 710054, China; 3School of Safety Science and Engineering, Xi’an University of Science and Technology, Xi’an 710054, China

**Keywords:** brain fatigue, video terminal display operation, sustained attention, event-related potential

## Abstract

Intelligent mines require much more mental effort from visual display terminal (VDT) operators. Long periods of mental effort can easily result in operator fatigue, which further increases the possibility of operation error. Therefore, research into how brain fatigue affects the sustained attention of VDT operators in intelligent mines is important. The research methods were as follows: (1) Recruit 17 intelligent mine VDT operators as subjects. Select objective physiological indicators, such as reaction time, error rate, task duration, flicker fusion frequency, heart rate, electrodermal activity, and blink frequency, and combine these with the subjective Karolinska Sleepiness Scale to build a comprehensive brain fatigue evaluation system. (2) According to the fatigue-inducing experiment requirements, subjects are required to carry out mathematical operations in accordance with the rules during the presentation time, determine whether the results of the operations fall within the [7, 13] interval, and continue for 120 min to induce brain fatigue. (3) Perform the standard stimulus button response experiment of the sustained attention to response task, before and after brain fatigue, and compare each result. The results show that: (1) When the standard stimulus appeared in the EEG experiment, the amplitude of the early N100 component before and after brain fatigue was significantly different. When the bias stimulus appeared, the average amplitudes of the P200 component and the late positive component, before and after brain fatigue, were significantly different, suggesting that the brain fatigue of VDT workers in coal mines would reduce sustained attention; (2) After the 120 min of the continuous operation task, the subjects showed obvious brain fatigue. The objective brain fatigue was followed by an increase in reaction time, an increase in error rate, a decrease in flicker fusion frequency, an increase in heart rate, an increase in electrodermal current, an increase in the number of blinks, and a larger pupil diameter, and both the subjective and objective data indicated more significant changes in the subjects’ brain fatigue at the 45th and 90th min. The results of the study could provide insight into the reduction in operational efficiency and safety of VDT operators in intelligent mines due to brain fatigue and further enrich the research in the area of brain fatigue in VDT operations.

## 1. Introduction

With the development of information technology, the coal industry is entering the intelligent mine construction phase, and the technical framework is gradually evolving [1]. China’s intelligent mining operations reached 813 in 2021, an increase of 65% compared to 2020. Since 2016, 370,000 workers have been eliminated from coal mines countrywide due to intelligent construction. Digital and intelligent systems have transformed the previous tasks of personnel monitoring and manual operation in the control room to today’s monitoring and control via electronic displays, which is also known as VDT operation. VDT refers to a visual display terminal, such as a computer system or personal computer. VDT operation is a general term for processing information such as data, text, and graphics on various electronic screens [2]. Modern VDT operation reduces the physical requirements, while the mental and sustained attention demands on the operator are much higher [3,4]. As a result, VDT operators may experience decreased efficiency or operational errors, which rarely cause direct injury or death, but might result in risks, and risks that have a high likelihood of occurrence are more likely to result in accidents. [5] Chinese mines saw a total of 356 accidents and 503 fatalities in 2021, with VDT operation and monitoring faults accounting for 42% of these incidents. As one of the primary causes of VDT operational errors, fatigue [6] has received extensive attention from mine managers and scholars.

Physical fatigue, brain fatigue, and mixed fatigue are three different types of fatigue [7]. Brain fatigue has the greatest effect on VDT. When it comes to VDT in intelligent mines, for example, the shifts usually run as “three-eight” [8], which means that each of the three shifts consists of about three workers who alternately work continuously for eight hours each shift. The operator is required to attend to the various monitoring screens, alert announcements, information pop-ups, and other information in the dispatch monitoring room concerning prolonged monotonous monitoring tasks. As the duration of VDT work increases, the operator may experience visual and mental fatigue, and consequently, the related indices of attention (e.g., reaction time, error rate, etc.) may change [9,10,11,12,13]. In this case, the operators’ response and manipulation ability is usually decreased, their capacity to perceive individual changes decreases, their cognitive function is impaired, their capacity for sustained attention is affected, and their reaction time for making judgments regarding dangerous situations is lengthened. As a result, dangerous situations for underground workers and apparatus are not immediately addressed [14,15]. This shows that VDT operation in the dispatch monitoring room requires the operator to maintain sustained attention, and the resulting fatigue may have an impact on the operator’s ability to process and retrieve information, and mistakes will inevitably occur during operations, in this case [16,17]. Therefore, it is crucial to research how brain fatigue from VDT operations affects sustained attention in the context of coal mine intellectualization.

Although there have been many studies on brain fatigue assessment methods, there are currently no unified standards for assessing brain fatigue because each method and technique is different. The earliest method used to assess fatigue is the subjective assessment method, which consists mainly of questionnaires and fatigue scales. Commonly used scales include KSS, the Multidimensional Fatigue Inventory (MFI)-20, the NASA Task Load Index (NASA-TLX), and the Epworth Sleepiness Scale (ESS) [18,19,20]. These scales or questionnaires follow a similar approach, asking the subject to assign a value or a subjective score based on the content of the scale in relation to their own situation, namely to present their feelings and performance after fatigue in the form of a verbal description. The advantages of the subjective assessment method are simplicity, acceptability, non-intrusiveness, speed, and low financial cost of use. The disadvantages include the difficulty in agreeing on fatigue criteria due to individual causes and the possibility that subjects may deliberately conceal their true feelings in order to meet experimental expectations. The objective assessment method uses tools to monitor physiological, psychological, and biochemical indicators in real time and records the indicators of the human brain in different states of fatigue, which are then compared and analyzed to determine the degree of fatigue. This method can be subdivided into biochemical, psychological and behavioral, and physiological methods, depending on the indicators used in the assessment of fatigue.

In general, although brain fatigue has been extensively researched and assessed by numerous methods, there are only a few studies on brain fatigue in the context of intelligent coal mines for underground VDT operators, and the few that have been conducted have used a single method of measuring brain fatigue, and of these, even fewer have examined the effects of brain fatigue on the sustained attention of VDT operators. However, with the increase in intelligent mine control rooms, the workloads for VDT operators have increased dramatically, resulting in longer VDT work periods. The resulting brain fatigue can affect cognitive processes and cause operational accidents, which is potentially hazardous to the profession of continuous monitoring. Therefore, the purpose of this study is to determine whether brain fatigue brought on by VDT operations in intelligent mines affects sustained attention by selecting VDT operators with different positions in intelligent coal mining enterprises as subjects, and designing experiments that incorporate event-related potentials, subjectively using KSS and objectively using reaction time, error rate, flicker fusion frequency, physiological indicators (the number of eye-blinks, blink frequency, electrodermal activity, and heart rate variability) to comprehensively evaluate brain fatigue in VDT operators to ensure the authenticity and validity of the subsequent EEG signal data analysis, and then combine these with the event-related potential technique to deeply explore the effect of brain fatigue on sustained attention.

## 2. Materials and Methods

In this study, 120 min CPT is applied to simulate VDT operations in intelligent coal mining enterprises to induce brain fatigue in operators. The first monitoring and recording of the EEG is performed for the sustained attention to response task (SART) before the onset of brain fatigue, and the second monitoring and recording of the EEG is conducted after the onset of brain fatigue until the end of the experiment, using a designed general experimental task of sustained attention as the stimulus sequence procedure for the ERP experiments. The physiological data of the subjects are collected simultaneously by Tobii telemetry, EEG, and physiological sensors during the experiment, while the behavioral data is recorded in real time in the background. After the experiments were completed, the collected signals were pre-processed and analyzed offline after all subjects had successfully completed the experimental data collection to obtain the subjects’ eye movement indicators, ERP averages, and EEG topography to investigate the differences in their sustained attention before and after brain fatigue. The CPT task was selected as the fatigue-inducing task for VDT operators, and each subject was required to perform the 120 min CPT task.

### 2.1. Experimental Equipment and Environment

The experimental site was chosen as a vacant office on the 3rd floor of the Huaneng Qingyang Coal power Co. (Qingyang, China) office building in an intelligent mine, where the temperature, humidity, and light levels were moderately regulated. In order to ensure a good experimental environment, the experimental process was strictly controlled for extraneous factors, the experimental procedures were strictly followed, extraneous people were prevented from interfering with the experimental process, and silence was maintained in the indoor and outdoor working environment to prevent noise from affecting the subjects. The experiment took 14 days to complete.

For the experimental equipment, an ASUS ROG laptop was used for the simultaneous acquisition of subject data from EEG and eye movement signals using the ErgoLAB human–computer environment synchronization platform, synchronized with a Lenovo 19″ HD widescreen LED computer terminal monitor with a resolution of 1920 × 1080 pixels.

### 2.2. Subject Selection

For the results of the study to be valid and to remove bias arising from gender, the subjects engaging in the experiment must be representative. Based on a comprehensive consideration of age, gender, intellectual and physical health factors, etc., we chose to recruit dispatching room staff from the Huaneng Qingyang Coal power Co, whose daily work and study involve prolonged exposure to electronic screens and require cognitive effort and exertion during operation, making them typical of VDT operators.

According to the ERP test specifications, a repeated test method is used, with the number of participants below 20, and multiple responses are recorded for each sample, thus ensuring a volume of data [21]. Therefore, 17 subjects are recruited for the experiment, including 9 males and 8 females. A total of 6 subjects are recruited for the monitoring position, 8 subjects for the dispatching position, and 3 subjects for the hoisting room position, and participants between the ages of 25 and 45 are recruited as subjects. The subjects selected satisfy the standards for the eye movement and EEG tests and are free of any sleep-related disorders. Precautions are taken in advance to minimize the effect of individual differences and irrelevant factors on the subjects’ brain fatigue status. All subjects enter voluntarily in the experiment, and a fee is paid to the subjects at the end of the experiment. The experiment complies with the specific principles and requirements of the relevant ethical and moral codes, and strictly adheres to the relevant safety procedures.

### 2.3. Experimental Materials

#### 2.3.1. Karolinska Sleepiness Scale (KSS)

Combining the KSS (subjective indicator), the reaction time, and the EEG signal (objective indicators) to measure the sustained attention is relatively scientific and reasonable [22]. Therefore, the KSS was used in this experiment as a subjective measure to record the subjective feelings and brain fatigue state of the subjects. The KSS includes 10 levels of sleepiness, and the matching scores range from 1 to 10. The scale stipulates that a higher score indicates greater subjective brain fatigue, and conversely, a lower score indicates lower subjective brain fatigue. In the experimental design, subjects were asked to fill out the scale in two separate sessions, before and after the brain fatigue experiment, based on their most realistic feelings, and the results were statistically analyzed at the end of the experiment.

#### 2.3.2. EEG Experimental Materials

The sustained attention to response task (SART) is a better measure of sustained attention than several different instruments of sustained attention, such as the traditionally formatted vigilance task (TFT) [23]. Therefore, SART was chosen for the EEG experiment, and the whole experimental sequence process was written and presented using the ErgoLAB platform.

The stimulus material consisted of ten numbers from 1 to 9 (set the number format to Arial font, font size 48, white font presented in the center of the screen with a black background); the number 3 was the deviant stimulus, with 20% probability of occurrence, the remaining nine numbers (1, 2, 4, 5, 6, 7, 8, 9, 0) were the standard stimuli accounting for 80%, and all numbers appeared randomly in proportion. Subjects responded by pressing ‘P’ when the standard stimulus appeared, and not when the deviant stimulus (number 3) appeared.

Subjects enter the experiment according to the cues, and the experiment is divided into a formal and a practice section. The formal part of the experiment consists of 2 blocks, with 180 trails in each block.

Each number (stimuli) was presented for 400 ms, and the stimuli interval was 1100 ms. Subjects had to respond to a keystroke as soon as the number appeared to proceed to the next trail, with a maximum of 1500 ms allowed for the keystroke response, and if there was no keystroke response, the next stimuli would automatically follow after 1500 ms. Figure 1 shows the stimulus presentation sequence of the SART paradigm in the EEG experiment.

#### 2.3.3. Stimulus Material for Brain Fatigue Induction

The continuous performance test (CPT) was used to induce brain fatigue in this experiment. CPT is also widely used to measure sustained attention, and a typical CPT presents a sequence of stimuli that requires the subject to maintain sustained attention while responding as soon as possible to a pre-set target stimulus [24]. The stimulus interface program for this experiment is programmed using E-prime software, which includes functions for experimental design, generation, manipulation, data collection, editing, and pre-processing for analysis. It also provides comprehensive timing information and event details (including presentation time and response time) for further analysis of relevant data, estimating how long the main test will take to run. Figure 2 shows a screenshot of the interface for programming CPT tasks using E-prime.

As human–computer interactive VDT operations in intelligent mines are normally performed in a seated position, while continuously looking at the display and repeatedly using the dominant hand to manipulate the input device to complete the VDT operation, in order to simulate the brain resource consumption of daily VDT operations, the CPT task was designed by E-prime programming, with these specifications in mind, and each subject was required to perform a 120 min CPT task to induce brain fatigue, as shown in Table 1.

There were 75 trails in each of the 8 15 min blocks that made up the 120 min CPT task. Each trail was presented for 2000 ms from the gaze point “+” (to eliminate the effect of position on the visual search) [25]. Following the presentation of the trails, a set of random stimulus numbers was shown on the display from left to right, each set consisting of three random numbers between 0 and 9, with the presentation time of 5000 ms for each set. During the presentation time, the subject was asked to complete a mathematical operation according to the rules, and the result of the operation was recorded as N4. If the subject believed that the result of the operation N4 belonged to the interval [7, 13], he/she should swiftly press P on the keyboard, while if the subject believed that the result of the operation N4 did not belong to the interval [7, 13], he/she should swiftly press Q on the keyboard. The experimental steps are shown in the Figure 3.

### 2.4. Experimental Process

The experimental sequence of the EEG experiment and the brain fatigue induction experiment (CPT task) is shown in Figure 4.

To ensure that the experiment is successfully conducted, the subjects receive a thorough explanation of the experimental procedures and precautions, as well as the use of the EEG, physiological recorder, and oculomotor, and read and sign a written informed consent form before proceeding to the preparation and formal stages of the experiment. Each subject received appropriate compensation at the end of the experiment.

After preparing the various experimental equipment and controlling the experimental environment prior to the start of the experiment, the following formal experiment of approximately 3 h in length begins:

Step 1: Subjects fill in basic personal information and sit quietly at the lab bench for 10 min to adjust their personal state.

Step 2: Subjects are instructed to select options on KSS according to their current subjective fatigue state.

Step 3: Subjects are instructed to make 3 observations of the flicker fusion frequency meter at the same parameter setting and record the final mean value.

Step 4: Experimenters debug the equipment to ensure that the EEG and physiological sensors are working stably and transmitting real-time data efficiently on the Ergolab human–machine environment synchronization platform. While the experiment is ongoing, subjects are asked to adjust their sitting position to ensure that the Tobbi eye-tracking device can record their eye movement data.

Step 5: As soon as everything is ready, subjects can begin practicing the experiment before it formally starts. After the completion of the pre-brain fatigue EEG experiment, the experimental apparatus is organized and data is collected.

Step 6: Following a brief rest, subjects participate in the CPT task for the next 120 min, in good physical and mental condition, for brain fatigue induction, filling in the KSS and measuring the flicker fusion frequency every 15 min, 8 times in total. Prior to beginning the task, subjects must read the task interface and introduction carefully and perform a 2 min adaptation exercise.

Step 7: The brain fatigue induction experiment ends at this point.

Step 8: The post-fatigue EEG experiment starts immediately. The post-fatigue EEG experiment follows the same procedure as the pre-fatigue EEG experiment, repeating steps 2 through 5.

Once the experiment is finished, the collected data are collated and summarized for later offline data analysis. The differences and patterns of change are analyzed by comparing the subjective scale and the physiological data of the subjects before and after the 2 h CPT task-induced brain fatigue.

Figure 5 shows the subject being set up with an EEG cap on his head, a heart rate collector on his right arm, and an electrodermal collector on his left wrist. While the subject is undergoing the EEG experiment, the researcher sits quietly by and monitors whether the real-time data is being recorded correctly and efficiently, as shown in Figure 6.

## 3. Results

### 3.1. Subjective Scale Data Analysis of Brain Fatigue

When engaging in prolonged, monotonous, memory-demanding brain activities, people gradually experience varying degrees of brain fatigue over time, such as wandering, difficulty focusing, and sleepiness. Therefore, the subjective feelings of the subjects in the experiment serve as references for the detection of brain fatigue.

The KSS was used in the brain fatigue experiment to assess the subjects’ subjective brain fatigue perceptions and status. The differences between the KSS scores taken before and after brain fatigue are displayed in Table 2.

The subjective questionnaire scores were divided into two groups, pre-brain fatigue and brain fatigue, and the data from both groups were subjected to paired *t*-tests using SPSS 26.0; the test results are shown in Table 3.

The magnitude changes of mean scores before and after brain fatigue are shown in Table 4.

It is evident from Table 3 and Table 4 that: (i) the mean scores of post brain fatigue increase compared to pre-brain fatigue. The higher the score of KSS, the higher the degree of brain fatigue. This shows that the subjective brain fatigue level of the subjects increased following the brain fatigue experiment; (ii) a paired-samples *t*-test was conducted on the mean scores of KSS, and *p* = 0.000 < 0.01 was obtained, which is a significant difference. The above are the results of the subjective statistics, including great uncertainty. The following objective physiological indicators were used to more accurately evaluate the subjects’ brain fatigue status.

### 3.2. Objective Data Analysis of Brain Fatigue

#### 3.2.1. Reaction Time and Accuracy Rate

Reaction times are divided into two categories: simple reaction time and discriminative reaction time. The former refers to the time required for the subject to respond to a single stimulus, and the latter refers to the time required for the subject to respond to the selection of a stimulus that matches the required response.

Figure 7 and Figure 8 show the reaction time and accuracy rate for each of the 17 subjects. In this experiment, the reaction time test used for the two EEG experiments before and after brain fatigue employed discriminative reaction time, and each subject’s reaction time was recorded for both phases of the EEG experiment. A total of 360 responses were recorded on the screen, 288 of which required a discriminative response, and the 288 response times of each of the 17 subjects were averaged for analysis. A percentage of the correct responses out of 360 is used to analyze the accuracy rate.

The results shown in Table 5 and Table 6 reveal significant differences between the subjects’ reaction time and accuracy rate before and after brain fatigue, with subjects taking an additional 43 ms to complete the same task after brain fatigue was induced, and with a 5.3% decrease in accuracy rate. The subjects’ accuracy rate and mean correct reaction time under the same task before and after brain fatigue were statistically analyzed, and a paired-samples *t*-test was conducted for each sustained attention to response task. The results indicate that there was a significant difference in the mean correct reaction time before and after brain fatigue (t = −3.512, *p* = 0.04 < 0.05), with a statistically significant difference in the accuracy rate between the two (t = 2.407, *p* < 0.05), suggesting that as the duration of VDT operations increases, the subjects’ memory load rises and their thinking time lengthens, resulting in longer reaction times and a decrease in accuracy rate.

#### 3.2.2. Flicker Fusion Frequency

The critical flicker frequency (CFF) is the minimum frequency at which a person viewing a flickering spot of light may precisely create the stimulus that results in the sensation of flash fusion in the human eye. The settings in this experiment are: fixed green spot, fixed duty cycle 1:1, fixed spot brightness 1:8, fixed background brightness 1:4, and a continuously adjustable flicker frequency of 20 Hz~60 Hz. Each subject measured the current flicker fusion frequency with the highest accuracy, according to the specified test method. The flicker fusion frequency was averaged using three measurements to reduce data bias. Table 7 shows the flicker fusion frequency values recorded for each of the 17 subjects. Figure 9 shows a comparison of the flicker fusion frequency values before and after brain fatigue.

The degree of human brain fatigue can be reflected by the measured value of CFF. When the human body develops brain fatigue, the flicker fusion frequency threshold declines, with a change in threshold of 1.0 to 3.9 Hz for mild fatigue; 4.0 to 7.9 Hz for moderate fatigue; and above 8.0 Hz for deep fatigue. Table 8 shows the results of the paired samples *t*-test for the mean scores of the flicker fusion frequency meter before and after fatigue. As can be seen in Table 8, the total mean difference in flicker fusion frequency before and after brain fatigue induction reached 4.49 Hz, and the subjects reached moderate fatigue after brain fatigue induction.

#### 3.2.3. Heart Rate and Dermal Electric Variation

Heart Rate Variation

Mean HR, or the average heart rate, which represents the number of beats per minute, was used as the heart rate variability indicator in the experiment. The normal heart rate for adults ranges from 60 beats/min to 100 beats/min. Figure 10 shows the variation in heart rate values before and after the subject’s brain fatigue.

The heart rate data were divided into two groups according to pre- and post brain fatigue, and a paired samples *t*-test was performed on the two groups using SPSS 26.0. The results of the test are shown in Table 9.

Overall, it seems that the heart rate rises as the subjects’ brain fatigue level increases. The result of *p* = 0.000 < 0.01 in the paired samples *t*-test indicates that the subjects’ heart rates vary significantly before and after their brain fatigue.

Dermal Electric Variation

SC, or skin conductance, is utilized in the experiment as the indicator of the electrical signal properties of the skin, and its unit is μΩ. When a person is exposed to brain fatigue, stimulus conditions, or emotional changes, the skin resistance decreases and the electrical skin signal increases. Therefore, the subject’s brain fatigue state can also be effectively reflected by the electrical skin response signal as an indicator. Table 10 shows the KSS scores for each of the 17 subjects. Figure 11 shows the average skin electrical values of the subjects before and after brain fatigue.

The results of the test are shown in Table 11. A paired-samples *t*-test of the mean electrodermal values before and after the subjects’ brain fatigue induction revealed a significant difference (t = 3.4, *p* < 0.05) with sustained VDT operations, giving the subjects significantly higher electrodermal values after brain fatigue than before brain fatigue.

#### 3.2.4. Eye Tracking Data

Combining the status of eye movement indicators associated with brain fatigue and the eye movement behavior recorded by the Tobii Pro Nano eye-tracking device, two eye-movement indicators are obtained from this study: pupil diameter and frequency of blinks. Based on the recorded data, the mean values of the above indicators are calculated for the participants before and after fatigue induction, as shown in Figure 10.

Figure 12 shows that: (i) the mean pupil diameter of the subjects is smaller after brain fatigue than that before brain fatigue; (ii) the average number of eye blinks during the task period is higher than that before brain fatigue, suggesting that brain fatigue increases blink frequency.

When subjects blink more frequently after brain fatigue than they do before brain fatigue, it suggests that the subject’s attention to the current task has been shifted internally and sustained attention has decreased [26]; moreover, blinking also causes activation of brain areas that process external visual stimuli, i.e., blinking decreases the processing of visual information [27].

These two indicators provide a lateral insight into the brain fatigue state of the operator during VDT tasks. The data also provides good validation that blinking can reflect a person’s brain fatigue and psychological state. When the person is more focused, both the blink duration and the blink frequency decrease, while an increase in the blink frequency indicates the state of brain fatigue and poorer concentration. Changes in pupil diameter can also be a good indicator of brain fatigue, with a strong correlation between pupil size and brain fatigue.

### 3.3. EEG Data Analysis

#### 3.3.1. ERP Analysis

At the end of the experiment, the data were first preprocessed, and among the 17 EEG data, 3 datasets with disturbed wave amplitudes that could not be used for subsequent analysis were excluded; the components generated by 4032 (288×14) standard stimuli and 1008 (72 × 14) deviant stimuli were then superimposed on the remaining 14 subjects, before and after the fatigue induction, to obtain the waveforms of the event-related potentials before and after the fatigue. Based on the grand mean maps of ERP obtained by pre-processing the EEG data, the EEG indicators selected for analysis in this study are: N1 waves in the anterior part of the scalp between 80 ms and 120 ms, N1 waves in the posterior part of the scalp between 150 ms and 200 ms, P2 waves in the posterior part of the scalp between 190 ms and 260 ms, N2 waves in the posterior part of the scalp between 260 ms and 400 ms, and LPC late waves in the anterior part of the scalp between 400 ms and 600 ms.

Based on the extraction and processing of EEG data, the mean wave amplitudes of EEG indicators were analyzed in paired samples for different states of brain fatigue (pre-brain fatigue and post-brain fatigue), which were used to investigate the effects of the subjects’ sustained attention to the same task and the changes to each EEG indicator before and after the state of brain fatigue.

N1 waves induced by standard stimuli

During the time window of 100 ms to 180 ms after the presentation of the standard stimulus, according to the ERP total mean map, the N1 wave was observed in the forehead zone, the central zone, and was more pronounced at five electrode points: FC1, FC2, CZ, CP1, and CP2; therefore, the brain signals from these five leads were selected for observation and demonstration. The total mean EEG waveforms induced at these five leads are shown in Figure 13, with a time window of 1000 ms, for analysis, where black represents the event-related potential waveform measured before brain fatigue was induced, and red represents the event-related potential waveform measured after brain fatigue was induced. It can be seen that the standard stimulus evoked a more pronounced N100 waveform, and that the mean waveform amplitude was significantly lower in the post-brain fatigue than that in the pre-brain fatigue group.

N1, P2, N2, LPC late waves under deviant stimuli

During the time windows of 150~200 ms, 200~260 ms, 260~400 ms, and 400~600 ms after the presentation of deviant stimuli, the N1, P2, N2, and LPC waves are observed in the frontal and central area, according to the grand mean maps of ERP. These waves are more pronounced at the three electrode points: FC1, FC2, and CZ; therefore, that brain signals from these three leads are selected for display. The total mean EEG waveforms induced at these three leads are shown in Figure 12, with an analysis time window of 1000 ms, where black represents the event-related potential waveform measured before the brain fatigue induction, and red represents the event-related potential waveform measured after the brain fatigue induction.

Figure 14 shows that the general direction and time axis of the waveforms of each component, before and after brain fatigue, are very similar. The highest point of the positive peak was reached at around 200 ms, and the average wave amplitude after brain fatigue increased compared with that before brain fatigue; the highest point of the negative peak was reached at around 250~300 ms for N200 induced by deviant stimulation, and the late LPC wave was also induced by deviant stimulation at 400~600 ms; at the same time, some differences in the wave amplitudes of N1, P2, N2, and LPC, before and after brain fatigue, could be seen. To further test whether this difference is significant, the wave amplitudes of the standard and deviant stimuli are counted and then tested by paired sample analysis, with brain fatigue state (pre-brain fatigue and post-brain fatigue) as the independent variable and the wave amplitudes of N1, P2, N2, and LPC as the dependent variables; the results are shown in Table 12.

Furthermore, as shown in Table 12, there is a significant difference between the mean N100 wave amplitude evoked by the pre-brain fatigue state and the N100 wave amplitude evoked by the post-brain fatigue state under the standard stimulus (*p* = 0.035 < 0.05); but there was no significant difference between the two under the deviant stimulus (*p* = 0.538 > 0.05). In addition, with deviant stimuli, the P200 mean wave amplitude evoked by the pre-fatigue state differs significantly from that evoked by the post-fatigue state (*p* = 0.038 < 0.05); the N200 mean wave amplitude evoked by the pre-fatigue state differs insignificantly from that evoked by the post-fatigue state (*p* = 0.259 > 0.05); the LPC late wave amplitude evoked by the pre-fatigue state differs significantly from that evoked by the post-fatigue state (*p* = 0.0 < 0.05).

#### 3.3.2. EEG Topographical Maps

Figure 15 shows the EEG topographical maps using Matlab software and within the time window of 130~180 ms, that is, the N1 waveform components induced by standard stimuli in both pre-brain fatigue and post-brain fatigue states. From the brain topographical maps, it is observed that when completing the task in the pre-brain fatigue state, the subject’s anterior brain region is activated, most obviously in the central area; whereas, when completing the same task in the post-brain fatigue state, both the amplitude and the activation in the central area are significantly lower than in the pre-fatigue state, and the temporal area is also activated.

Figure 16 shows the EEG topography during the time windows of 80~120 ms, 190~260 ms, 260~400 ms, and 400~600 ms when the deviant stimuli appeared, that is, the N1, N2, P2, and LPC late waveform components induced by the pre-brain fatigue and post-brain fatigue states in response to the deviant stimuli. The topography of the brain shows that in response to the deviant stimulus, the N1 component appears in the frontal-central and temporal lobes around 80–120 ms in the pre-fatigue state, the P2 component appears in the frontal region during the 190~260 ms time window, and the LPC late waves appear and gradually expand to most of the frontal and central area as time progresses beyond 400 ms, before disappearing. In the post-fatigue state, the N1 wave amplitude in the frontal and central area is lower in the 80–120 ms time window, but not significantly, while the P2 wave is active in the central region in the 190–260 ms time window and is more activated than before brain fatigue; the activity of the LPC late waves in the 400–600 ms time window is significantly lower than that before brain fatigue.

## 4. Discussion

In the experiment, accuracy rates, omission rates, reaction times, subjective KSS, and objective flicker fusion frequency meter data are recorded for all subjects performing the two-hour CPT (continuous operational task), as shown in Table 13. Subjects are required to maintain a high level of attention for cognitive continuous VDT operations during the 120 min CPT task, and the values recorded in the table are the mean values for all subjects at 15 min intervals.

As seen in Table 13, as the duration of the CPT task increases, the subjects’ reaction time lengthens, the accuracy rate decreases, the omission rate increases, and the flicker fusion frequency values and KSS scores show an incremental trend. As seen in Figure 17, after a CPT task lasting 45 min, the mean value of the subjects’ reaction time is 1683 ms, 224 ms longer than the mean value of 1459 ms at 30 min, while their accuracy rate dropped by 2.59%, from 94.72% to 92.13%; after a 90 min CPT task, the mean value of the subjects’ reaction time is 2095 ms, 273 ms longer than the mean value of 1822 ms at 75 min, while the accuracy rate dropped by 4.21%, from 91.28% to 87.07%. These two time periods show a more pronounced inflection point throughout the 120 min CPT task, and each time period is 45 min apart. As seen in Figure 18, the KSS scores of the 17 subjects at different times reveal that the mean scale scores follow a rising trend as duration increases; however, for each subject, there is no change in the scores of 6 subjects at the 30th minute, but there is a change in the scores of all subjects at the 45th minute. Therefore, the time at which the brain first enters potential fatigue may exist in the 30–45 min interval, the precise reasons for which require further research.

From the results of the ERP analysis, the wave amplitude of the N100 component involved in the early perceptual stages of cognitive processing in the presence of standard stimuli differed significantly between pre- and post-brain fatigue, with the N100 after brain fatigue significantly lower than before brain fatigue. As an exogenous EEG component, N100 is closely related to the brain’s cognitive processing of the physical properties of external information [28]. It primarily reflects the fact that early visual processing is based on attentional selection of objects. Using auditory stimulus elicitation, for example, a significant reduction in N100 wave amplitude was found as early as 1991, suggesting that subjects’ attention left the current task [29], and that N1 facilitates the processing of task-relevant ideas [30]. However, a reduction in N1 wave amplitude would imply a decrease in the facilitation of task-related thoughts. The significant decrease in N1 amplitude following brain fatigue in the present study suggests that the attention required to process the same stimuli is shifted or deviated, and that brain fatigue inhibits the processing of external stimuli by attentional resources after sustained VDT operations, resulting in attenuation of attention and a decrease in sustained attention.

The increase in P2 amplitude is associated with attentional disengagement from the current stimulus [31], and the early stages of sleep are characterized by an increase in P2 amplitude [32]. The present study reveals that the wave amplitude of P2 is significantly greater after the onset of brain fatigue than before. The combination of brain topography shows that anterior brain regions are activated, most notably in the frontal and central areas, as well as in the bilateral temporal regions, which is consistent with previous studies. The P200 component is a post-processing component whose amplitude reflects the extent to which a person is investing mental factors, namely, the number of cognitive resources allocated [33]. A significant increase in P2 amplitude represents a stage in which the brain needs to invest more mental load, or to mobilize more brain resources, which corresponds to an increase in the cognitive function of selective and sustained attention, implying that subjects need to expend more effort to make correct decisions in response to deviant stimuli as they arise.

Combining the experimental results with previous experience in selecting a suitable time window, it is found that the N200 component of the deviant stimulus cue evokes a peak wave amplitude at around 300 ms latency, and there is no significant difference in its mean wave amplitude when analyzed.

When deviant stimuli emerge, there is a significant difference in the 400–800 ms time window, i.e., late positive component LPC, before and after brain fatigue (mean wave amplitude is lower after brain fatigue than before brain fatigue). The EEG topography shows that there is extensive activation in the occipital-temporal-parietal lobes in the pre-fatigue state; after brain fatigue, activation in the frontal and central areas is reduced. It has been proposed that LPC is an endogenous component similar to P3b [34]; the “whole-brain neural work area” model suggests that LPC is closely related to the top-down attention component of perceptual processing and represents early consciousness generation [35]; the reduction in mean LPC late wave amplitude also mirrors the reduction in top-down attentional components at stimulus onset and reduced stability of attention, implying a reduction in sustained attention following sustained VDT operation-induced brain fatigue in the subjects.

## 5. Conclusions

In this study, N100, P200, and LPC late wave components, along with EEG topography, are extracted before and after brain fatigue based on ERP technology by designing an experiment and recording the corresponding subjective scale data, objective physiological data, and behavioral data of each subject to investigate the changes in EEG event-related potentials triggered by different brain fatigue states. The followings are the study’s primary conclusions:(1)The eight indicators of reaction time, error rate, task duration, flicker fusion frequency, heart rate, electrodermal activity, blink frequency, and pupil diameter are sensitive to changes in cognitive VDT sustained operational brain fatigue, which could be combined with subjective scales as evaluation indicators for creating a comprehensive brain fatigue evaluation system.(2)The same experimental task’s ERP results demonstrate that N1, P200, N200, and LPC components are evoked, both before and after brain fatigue, and that the common brain regions associated with attentional activation are the frontal areas, especially the prefrontal, central, parietal, and temporal regions. The wave amplitude of the N100 component evoked by standard stimuli is significantly lower in the post-fatigue state than in the pre-fatigue state, showing a decrease in sustained attention; the wave amplitude of the P200 component evoked by deviant stimuli is significantly higher in the post-fatigue state than in the pre-fatigue state, showing that subjects need to expend more cognitive effort to maintain attentional stability; the amplitude of the LPC late wave is significantly lower, showing a decrease in sustained attentional stability. The significant decrease in LPC late wave amplitude also points to a decrease in sustained attentional stability. In other words, the sustained attention of VDT workers in intelligent mines decreases with the development of brain fatigue.(3)Brain fatigue is an important factor affecting sustained attention in cognitive VDT operations. As the work period progresses, the operator‘s brain load increases, resulting in a state of brain fatigue and a decrease in sustained attention and performance, and after the 45th and 90th minute of continuous work, both subjective and objective data suggest that the operator develops a more pronounced state of brain fatigue.

This paper combined subjective evaluation scales and objective physiological indicators with the behavioral data and designed an experiment to qualitatively and quantitatively investigate the issue of the effect of brain fatigue on sustained attention in VDT operators in intelligent mines. Although the early aim of the study is achieved, there are still many limitations, and the following areas can be explored in-depth in future research.

(1)The study’s objects were VDT operators in intelligent mines, but the subjects were limited to VDT operators in one intelligent mine, so it remains to be verified whether the results of the study are generally representative. The experimental findings will be tested in the future by applying them to real-life situations in other intelligent mines.(2)Due to the specific nature of the work and in order to control the variables in the experiment, a simulated work-induced brain fatigue experiment was adopted, but there are some differences between the simulated VDT work and the real work. If the data could be coupled with real working conditions for 24 h real-time testing, the results would be more credible.(3)Although the study is internally controlled, no control group is set up, suggesting that similar effects do not exist when similar duration activities are carried out. Subsequent experiments should be set up with suitable control groups to ensure the preciseness of the data.

## Figures and Tables

**Figure 1 ijerph-19-11034-f001:**
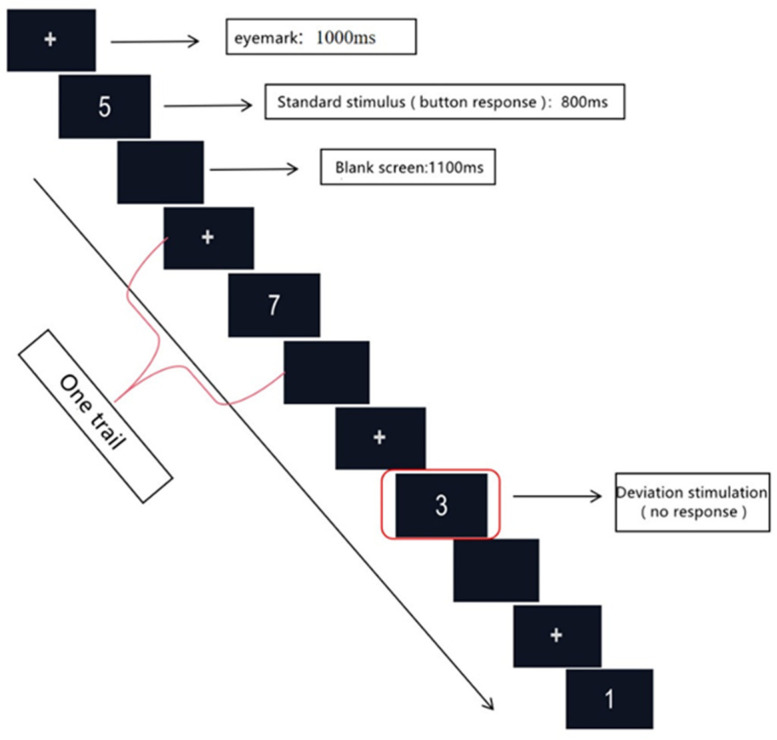
SART paradigm stimulus presentation sequence.

**Figure 2 ijerph-19-11034-f002:**
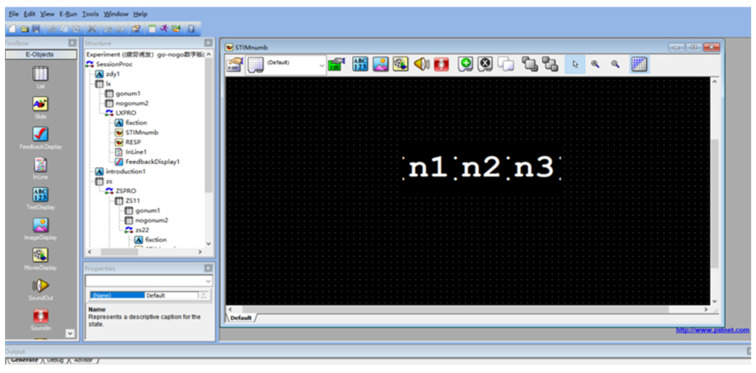
Task writing interface using E-prime software.

**Figure 3 ijerph-19-11034-f003:**
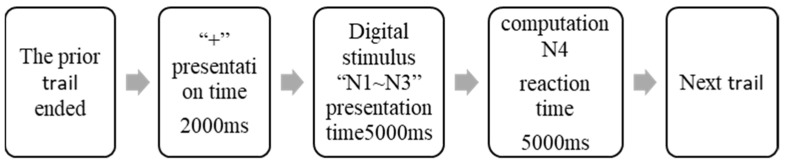
Single trail flow chart of CPT task.

**Figure 4 ijerph-19-11034-f004:**
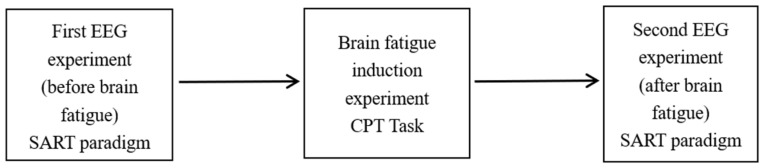
Flow chart of experimental sequence.

**Figure 5 ijerph-19-11034-f005:**
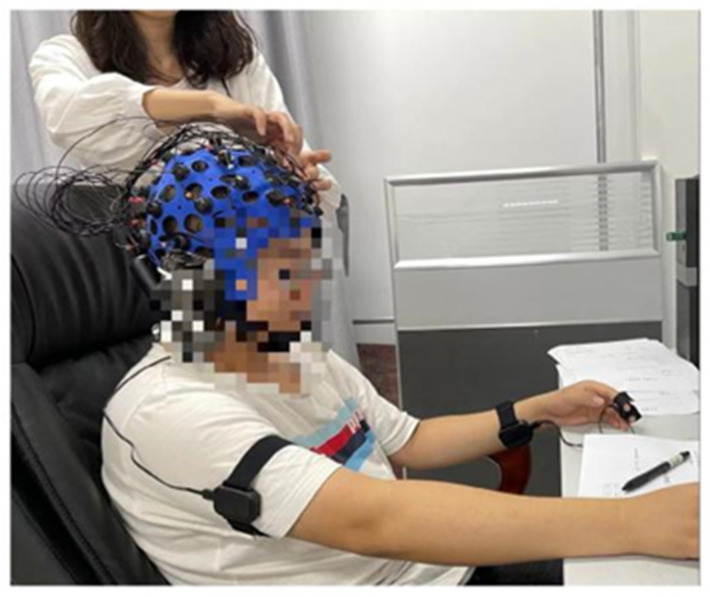
Testing machines.

**Figure 6 ijerph-19-11034-f006:**
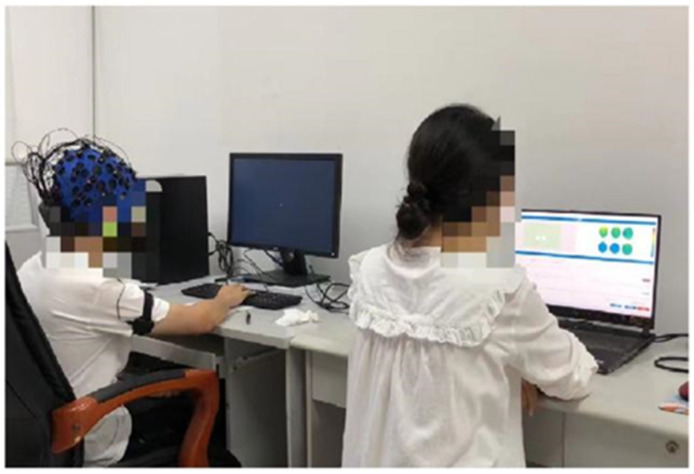
Experiment in progress.

**Figure 7 ijerph-19-11034-f007:**
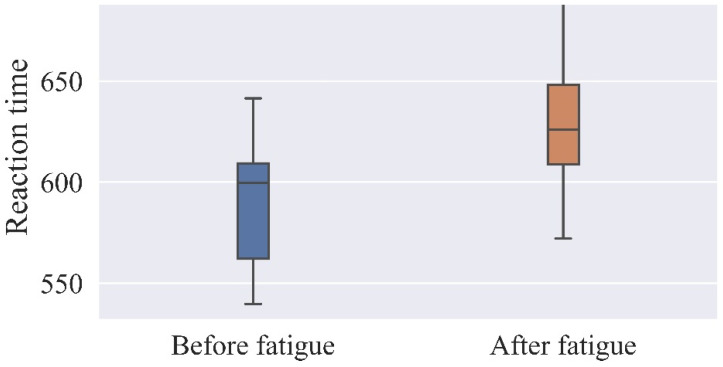
Reaction time before and after brain fatigue.

**Figure 8 ijerph-19-11034-f008:**
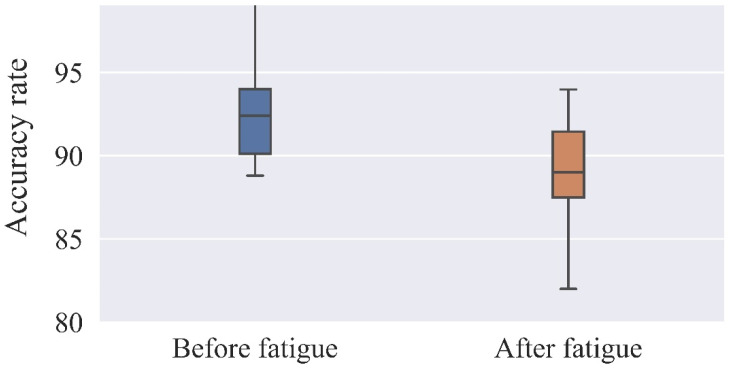
Accuracy rate before and after brain fatigue.

**Figure 9 ijerph-19-11034-f009:**
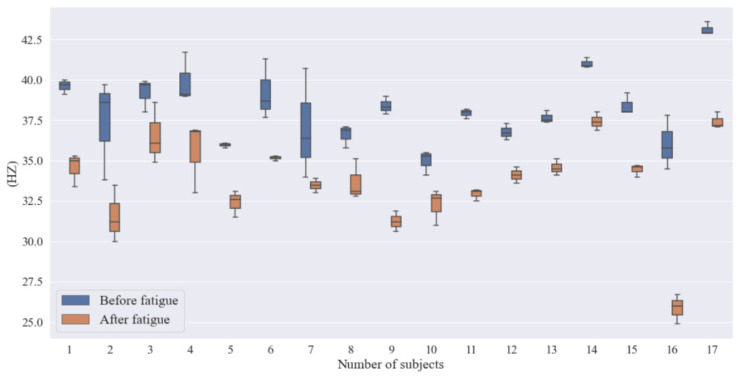
Flicker fusion frequency values before and after brain fatigue.

**Figure 10 ijerph-19-11034-f010:**
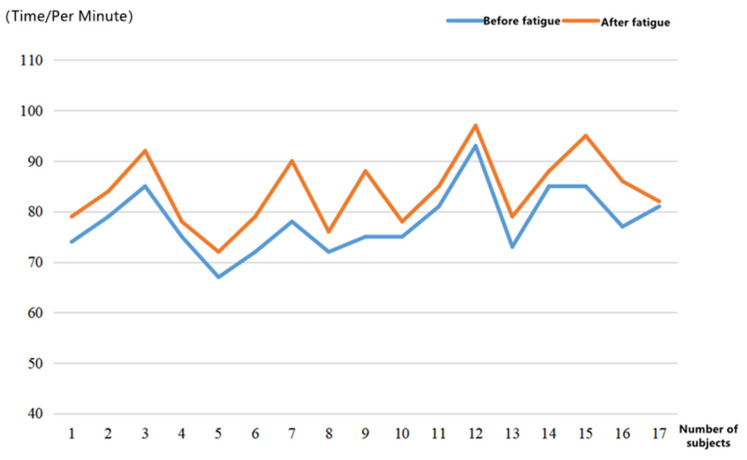
Changes in heart rate.

**Figure 11 ijerph-19-11034-f011:**
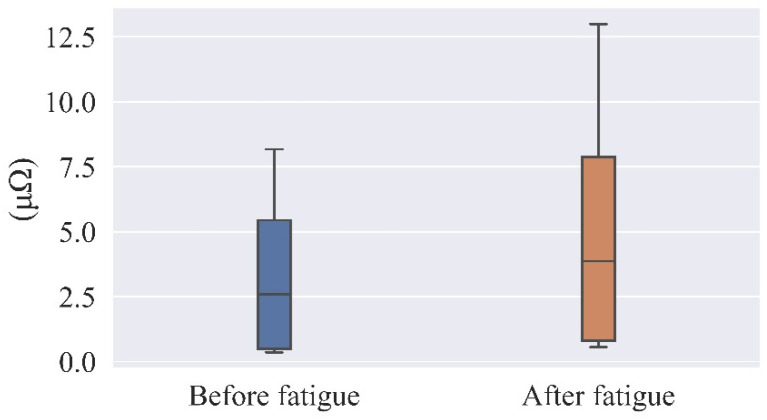
Comparison of mean values of subjects’ skin conductance before and after brain fatigue.

**Figure 12 ijerph-19-11034-f012:**
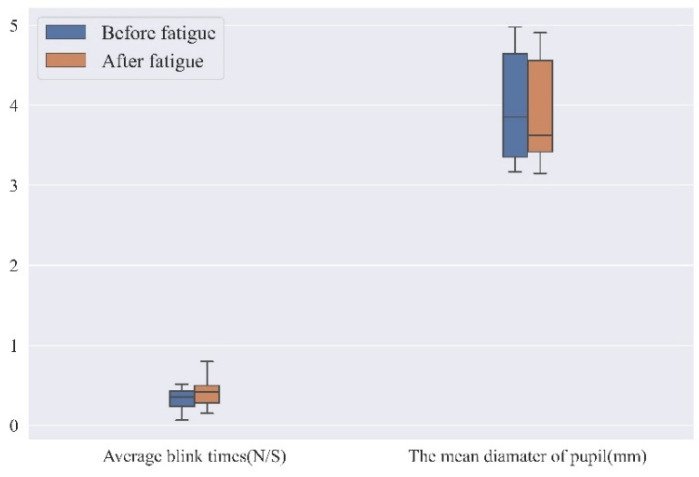
Mean eye movement index.

**Figure 13 ijerph-19-11034-f013:**
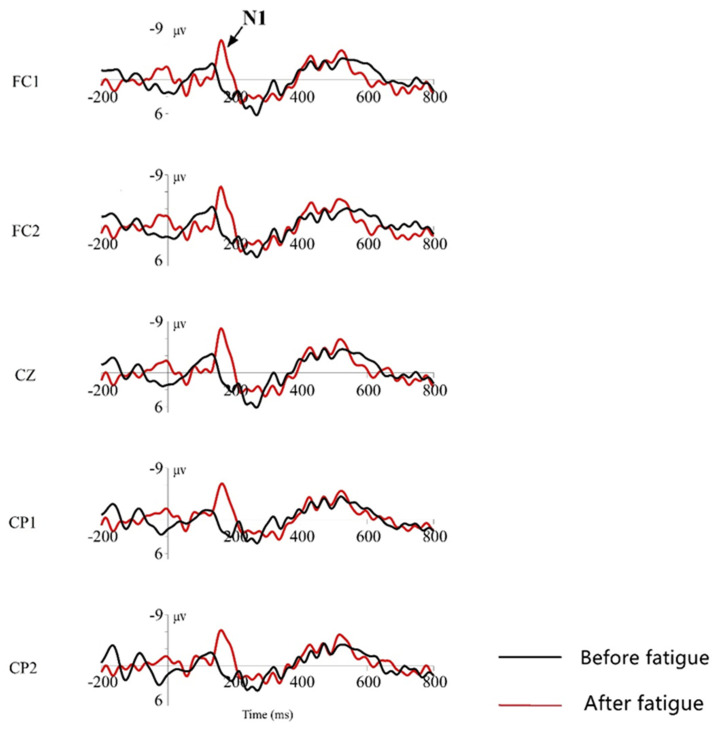
Total mean ERP diagrams at the FC1, FC2, CZ, CP1, and CP2 electrodes.

**Figure 14 ijerph-19-11034-f014:**
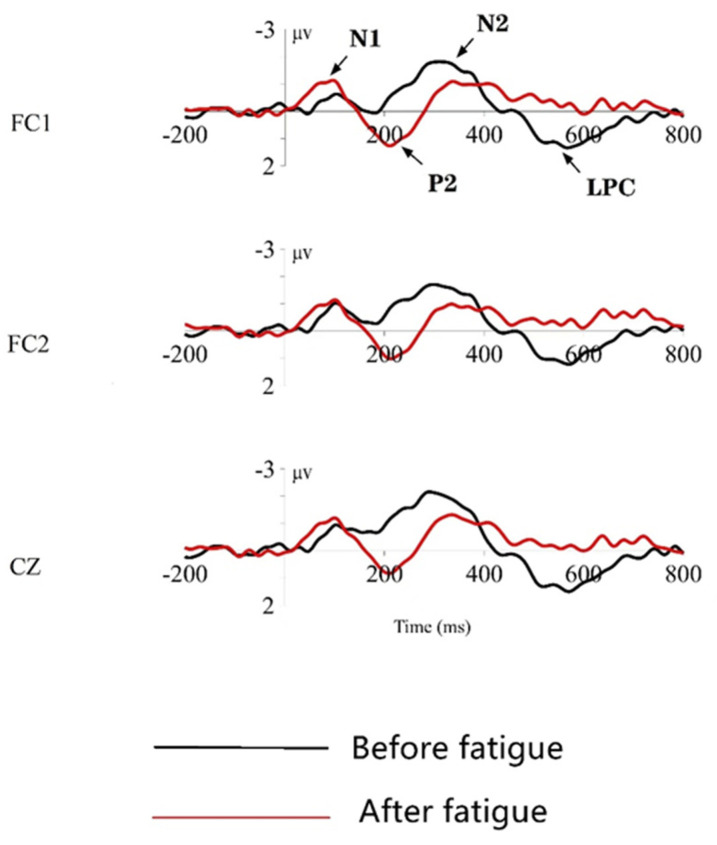
Total mean ERP plots/diagrams at the FC1, FC2, and CZ electrodes.

**Figure 15 ijerph-19-11034-f015:**
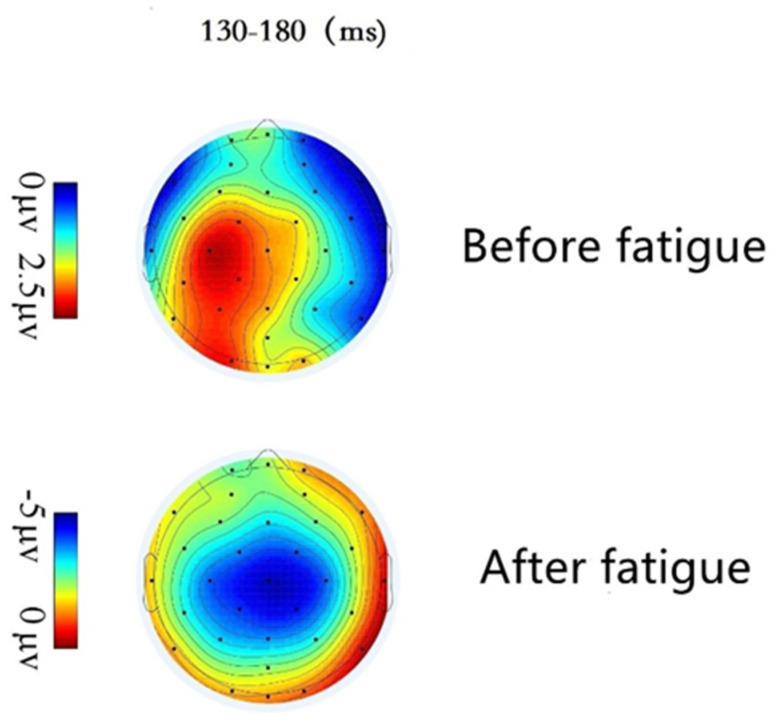
Brain topography in the 130–180 ms time window under standard stimulation.

**Figure 16 ijerph-19-11034-f016:**
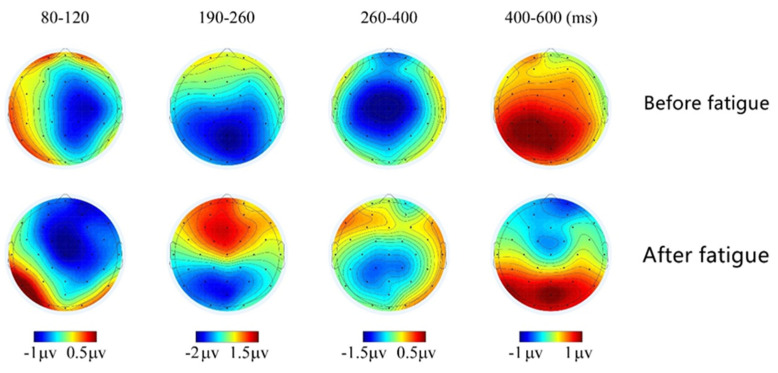
Brain topography at different time windows under deviant stimulation.

**Figure 17 ijerph-19-11034-f017:**
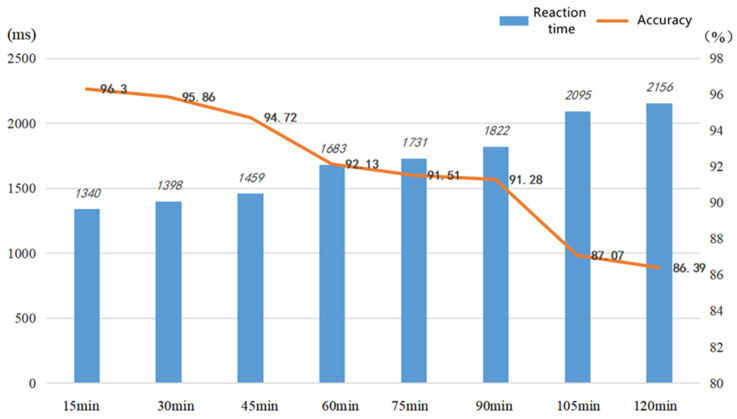
Mean values of response time and accuracy of subjects at different time periods for the CPT task.

**Figure 18 ijerph-19-11034-f018:**
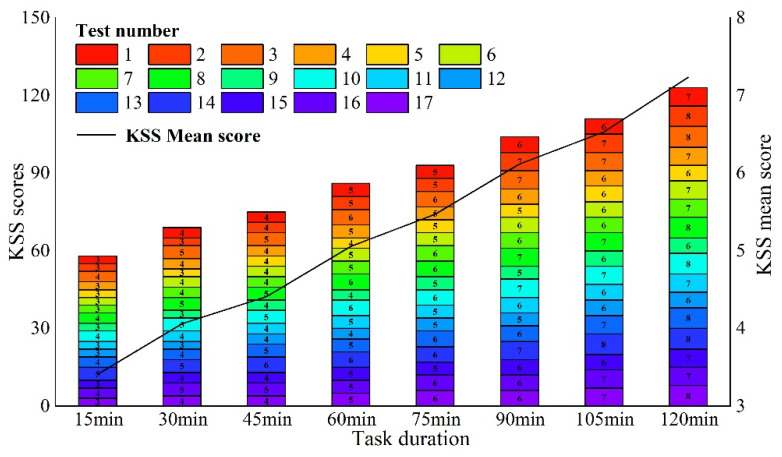
Scores and KSS means over time for the 17 subjects.

**Table 1 ijerph-19-11034-t001:** CPT Task Settings.

Continuous Performance Test (VDT)	Signal Rate	The Number of Trails
Operating conditions: N1 + N2 − N3 = N4, N4 ∈ [7, 13]	1:2	600
Response: Press P if in the interval; Press Q otherwise		
No response beyond 5000 ms: Miss		

**Table 2 ijerph-19-11034-t002:** KSS scores before and after brain fatigue.

Examinee	Before	After	Examinee	Before	After	Examinee	Before	After
1	3	6	7	3	6	13	3	8
2	3	7	8	4	8	14	5	8
3	3	6	9	3	7	15	3	7
4	3	7	10	3	7	16	4	8
5	3	8	11	3	7	17	3	8
6	3	7	12	3	7			

**Table 3 ijerph-19-11034-t003:** Paired samples *t*-test result.

	Mean	Std. Deviation	Std. Error Mean	t	df	Sig. (2-Tailed)
After fatigue–Before fatigue	−3.941	0.659	0.160	−24.671	16	0.000

**Table 4 ijerph-19-11034-t004:** Average score of KSS scale ($\bar{x}$ ± s).

Mental Fatigue State	Mean
Before fatigue	3.24 ± 0.562
After fatigue	7.18 ± 0.728

**Table 5 ijerph-19-11034-t005:** Paired samples *t*-test results for reaction time.

	Mean	Std. Deviation	Std. Error Mean	t	df	Sig. (2-Tailed)
After fatigue–Before fatigue	−42.451	49.832	12.086	−3.512	16	0.04

**Table 6 ijerph-19-11034-t006:** Paired samples *t*-test results for accuracy rate.

	Mean	Std. Deviation	Std. Error Mean	t	df	Sig. (2-Tailed)
After fatigue–Before fatigue	−42.261	49.620	12.034	2.407	16	0.029

**Table 7 ijerph-19-11034-t007:** Flicker fusion frequency values before and after brain fatigue.

Before Fatigue					After Fatigue				
Examinee	First (HZ)	Second (HZ)	Third (HZ)	Mean (HZ)	Examinee	First (HZ)	Second (HZ)	Third (HZ)	Mean (HZ)
1	39.10	39.70	40.00	39.60	1	33.40	35.00	35.30	34.57
2	38.60	33.80	39.70	36.75	2	31.20	33.50	30.00	31.57
3	39.70	38.00	39.90	39.20	3	36.10	38.60	34.90	36.53
4	39.10	39.00	41.70	40.35	4	33.00	36.80	36.90	35.57
5	36.00	35.80	36.10	35.97	5	33.10	32.60	31.50	32.40
6	38.70	41.30	37.70	39.50	6	35.00	35.30	35.20	35.17
7	36.40	40.70	34.00	37.03	7	33.00	33.90	33.50	33.47
8	36.90	35.80	37.10	36.45	8	32.80	33.10	35.10	33.67
9	38.30	39.00	37.90	38.40	9	31.90	31.20	30.60	31.23
10	34.10	35.50	35.30	35.40	10	33.10	32.70	31.00	32.27
11	38.20	38.00	37.60	37.93	11	33.20	32.50	33.10	32.93
12	36.70	36.30	37.30	36.80	12	34.10	33.60	34.60	34.10
13	37.50	37.40	38.10	37.67	13	35.10	34.50	34.10	34.57
14	41.40	40.80	40.90	40.85	14	37.40	38.00	36.90	37.43
15	38.00	39.20	38.00	38.40	15	34.60	34.70	34.00	34.43
16	37.80	35.80	34.50	36.03	16	24.90	26.70	26.00	25.87
17	43.60	42.90	42.90	43.13	17	37.10	37.20	38.00	37.43

**Table 8 ijerph-19-11034-t008:** Paired samples *t*-test result.

	Mean	Std. Deviation	Std. Error Mean	t	df	Sig. (2-Tailed)
After fatigue–Before fatigue	4.486	1.910	0.463	9.687	16	0.000

**Table 9 ijerph-19-11034-t009:** Paired samples *t*-test results of heart rate.

	Mean	Std. Deviation	Std. Error Mean	t	df	Sig. (2-Tailed)
After fatigue–Before fatigue	−5.941	3.344	0.811	−7.325	16	0.000

**Table 10 ijerph-19-11034-t010:** Skin conductance before and after brain fatigue.

Examinee	Before	After	Examinee	Before	After	Examinee	Before	After
1	7.88	9.17	7	0.49	0.56	13	7.42	7.88
2	4.47	10.69	8	1.14	0.65	14	5.45	6.63
3	6.35	10.05	9	0.47	0.81	15	3.28	3.87
4	8.18	13	10	2.23	3.73	16	2.59	4.07
5	0.49	1.33	11	0.39	0.67	17	4.27	5.67
6	0.36	0.75	12	2.06	3.26			

**Table 11 ijerph-19-11034-t011:** Paired samples *t*-test results of electrodermal values.

	Mean	Std. Deviation	Std. Error Mean	t	df	Sig. (2-Tailed)
After fatigue–Before fatigue	1.48647	1.78187	0.43217	3.427	16	0.003

**Table 12 ijerph-19-11034-t012:** Paired-sample test for mean wave amplitudes of N1, N2, P2, N2, and LPC, before and after brain fatigue.

		Paired Differences			t	df	Sig.
		Mean	Std. Deviation	Std. Error Mean			
Standard stimulus	N1(B-A)	−6.246	9.921	2.651	2.356	13	0.035
Deviant stimulation	N1(B-A)	0.265	1.567	0.419	0.633	13	0.538
	P2(B-A)	−1.766	2.867	0.766	−2.305	13	0.038
	N2(B-A)	−0.788	2.496	0.667	−1.181	13	0.259
	LPC(B-A)	1.029	0.648	0.173	5.942	13	0.000

**Table 13 ijerph-19-11034-t013:** Mean values of each component in the two-hour CPT task.

	15 min	30 min	45 min	60 min	75 min	90 min	105 min	120 min
Reaction time (ms)	1340	1398	1499	1623	1731	1822	2095	2156
Accuracy (%)	96.30	95.86	94.22	92.13	91.51	91.28	87.07	86.39
Missing report rate (%)	0.000	0.013	0.019	0.025	0.036	0.042	0.059	0.060
CFF (HZ)	38.19	37.70	37.21	36.69	36.03	35.41	33.98	33.70
Score of KSS scale (fraction)	3.41	4.06	4.41	5.06	5.47	6.12	6.53	7.24

## Data Availability

All relevant data used in the evaluation is displayed in the graphs contained within this article. Values for the raw data points are available upon request.

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
