# Peer review of "An Investigation of the Effects of Brain Fatigue on the Sustained Attention of Intelligent Coal Mine VDT Operators"

_ijerph, 2022, doi:10.3390/ijerph191711034_

Round 1
Reviewer 1 Report
The Investigation of the Effects of Brain Fatigue on Sustained Attention of Intelligent Coal Mine VDT Operators
The investigators studied the effect of brain fatigue in coal mine video terminal operators. It is a good study.
The abstract is not very clear and missing essential information as outlined below.
- How many subjects were involved in the study? While the study is internally controlled, it is recommended that authors use a set of reference subjects and show that similar effects are not there while performing similar duration sustained activity.
- It is unclear from the abstract what subjective and objective tests or analyses were performed on these subjects.
- Several short forms used in the abstract are not clearly explained. It is recommended to the authors that they explain the terms SART, N100, P200, LCP, etc., in the abstract before using them in the context of the abstract.
- The authors have mentioned comparative results. For example, how much significant difference, is it higher or lower, what is the p-value of the test, what test produced the result etc. It is good to have significant results but not helpful when they are not explained properly.
It is highly recommended that the authors reformat the abstract for better clarity.
Introduction:
- A brief introduction on the fatigue in video terminal operators would have been nice. The introduction has not provided any information on.
- How long is each shift?
- How many people usually work on these kinds of video terminals? What is the demographic of the operator population?
- What is the reported incidence rate, and how severe are the incidents?
- Establish the significance of the study not by providing the reference to previously published studies but instead by educating the audience.
- When the authors say "the operator may experience visual fatigue…" is this the only form of physical fatigue?
- "judgement of hazards is prolonged.". Do you mean the response time to make a judgment during a hazardous situation is prolonged?
- "Brain fatigue is defined as a tired mental state resul…" the paragraph seems repetitive and probably can be merged with the previous paragraph.
Methods:
- "6, 8 and 3 persons aged between 25 and 45 were recruited as subjects…" this line is unclear. Suggest breaking the statement.
- "Number 3 appearing was 20% and the remaining" is repeated several times. It does not have to be. After the first mention, it was clear how the stimulus was given.
- From the experimental explanation, it is not clear after how long of sustained monitoring activity the experiment started.
- The sections that provide an overview of the experimental setup and method to induce fatigue should be presented earlier in the methods section for better clarity.
- Figure 7 is a bit confusing. It is recommended to plot paired box plots to show the data individually or combined for all subjects.
- Figure 9, its recommended to show the box plots rather than bar plots for better clarity on the dataset.
The results are promising and well presented. However, there are a few points that authors should consider:
- Its easier to provide visual representation along with the number and statistics to better understand the results.
- The standard errors/ variance are given in many places, and many places are missing. It is recommended to use either standard error or variance, not both. Please also provide similar statistics for all measures.
- Several measurements seem to be time-dependent and taken for all individuals. It is recommended to provide the time-dependent changes.
- It appears that the authors selectively concealed some of the information. Even if the results are not promising or statistically insignificant, it is recommended to present them as it is for a fair judgment.
Author Response
Thanks for your advice.Please see the attachment.

Reviewer 2 Report
The paper presents sound statistical and procedures (e.g., fatigue measurement) relating to the effect of brain fatigue on coal mining operators. However, the following issues need to be resolved before publication.
The study involves human subject, but they author did not invoke and Institutional Review Board approval as per their statement. This is an enough justification for rejection, even though the methods and results warrant publication.
What rationale do the authors have that 17 subjects is enough to draw statistical conclusions?
The authors should not use abbreviations without definition in the abstract (e.g., SART).
Precisely define video terminal display at the beginning of the introduction.
Line 630, the paper has not been published yet.
Author Response
Thanks for your advice,Please see the attachment.

Round 2
Reviewer 1 Report
The authors have made all the necessary changes mentioned in the initial review.
I have only one concern. In some figures, the box plot's upper whisker is cut off. I am unsure if it is because of the track changes or if the figures were generated. If it was intentional, please ensure to include the whole image.